# A Contemporary Exploration of Traditional Indian Snake Envenomation Therapies

**DOI:** 10.3390/tropicalmed7060108

**Published:** 2022-06-16

**Authors:** Adwait M. Deshpande, K. Venkata Sastry, Satish B. Bhise

**Affiliations:** 1Sinhgad Institute of Pharmaceutical Sciences, 309/310, Kusgaon (BK), Lonavala 410401, India; satish@arogyalabh.org; 2Alliance Institute of Advanced Pharmaceutical & Health Sciences, Patel Nagar, Kukatpally, Hyderabad 500085, India; kvsastry_46@yahoo.com; 3Arogyalabh Foundation, Bibvewadi, Pune 411037, India

**Keywords:** snake envenomation, neglected tropical disease, plant-based antidote, traditional therapies, *Naja naja*, *Daboia russelii*, life saver

## Abstract

Snakebite being a quick progressing serious situation needs immediate and aggressive therapy. Snake venom antiserum is the only approved and effective treatment available, but for selected snake species only. The requirement of trained staff for administration and serum reactions make the therapy complicated. In tropical countries where snakebite incidence is high and healthcare facilities are limited, mortality and morbidities associated with snake envenomation are proportionately high. Traditional compilations of medical practitioners’ personal journals have wealth of plant-based snake venom antidotes. Relatively, very few plants or their extractives have been scientifically investigated for neutralization of snake venom or its components. None of these investigations presents enough evidence to initiate clinical testing of the agents. This review focuses on curating Indian traditional snake envenomation therapies, identifying plants involved and finding relevant evidence across modern literature to neutralize snake venom components. Traditional formulations, their method of preparation and dosing have been discussed along with the investigational approach in modern research and their possible outcomes. A safe and easily administrable small molecule of plant origin that would protect or limit the spread of venom and provide valuable time for the victim to reach the healthcare centre would be a great lifesaver.

## 1. Introduction

Snakebite envenomation is a fast advancing and serious situation, incomparable to any other acute disorder. The management of venomous snakebite is a complex process, and the victim’s physiological condition worsens quickly over time, demanding an aggressive treatment. Snake venom is a complex protein mixture, and every protein expresses its own biochemical activity. Few of the snake venom proteins are toxins while others are nontoxin proteins and enzymes. There is a great difference between the actions of the venoms of different snake species [1]. Neurotoxins are considered the most potent type of toxins as they quickly develop muscle paralysis and death occurs due to respiratory failure. Cytotoxins on the other hand induce haemorrhage and myonecrosis, causing severe local tissue damage at the bite site leading to permanent loss of organ function [1]. The nontoxin proteins from snake venoms induce various autopharmacological actions and may hamper the blood coagulation cascade. The other enzymes present in the snake venom act by inducing capillary damage leading to local tissue damage (hyaluronidases, phospholipases), various anti- and procoagulant actions (phospholipase A, proteinases) or induction of vasoconstriction and pain [2].

Snake envenomation has been classified as high-priority neglected tropical disease by the World Health Organization (WHO) [3,4]. A global initiative to reduce snakebite mortalities and morbidities to 50% by 2030 has also been launched [3,5]. In tropical countries, healthcare facilities in the rural areas might be scarce, and the victim might take significant time to reach the nearest facility. Reaching the nearest healthcare facility in the shortest possible time is of immense importance to avoid morbidities, permanent loss of function or death [6]. Once at the healthcare facility, the patient is critically observed for the vitals and treated accordingly with the snake venom antiserum (SVA) simply called antivenom. SVA is the only proven and recognized therapy for snake venom envenomation. First-aid measures such as immobilization of the bite site and restrictions on active movements of the victim are very important [7].

Before the discovery of SVA in the late nineteenth century [8] and its commercial availability later, snakebites were treated in various traditional ways that included physical therapies and mostly plant-based therapies. In the earlier days, practitioners of indigenous medicine in India used to maintain journals about their medical experiences. Mostly these journals were maintained secretly and passed to the next generation in the same household. Compilations of these journals were called *Aushadhi Baad*, meaning the medicinal collection. The practitioners would not disclose a particular therapy or an agent used therein to others. It was believed that certain therapies, if shared (with others), would lose their healing power [9], or the healer would lose their importance. The second reason appears to be more credible and would have resulted in the loss of several effective therapeutics to time.

In traditional Indian medicine (including *Ayurveda*), as most practised medicinal agents were derived from plant sources, books mentioning the use of plants in treating different ailments are available. These agents are known as *Vanaushadhi* (herbal medicine) in *Marathi*. Plant-based therapies provide clues for the discovery of newer molecules to a great extent. The terms *Gharguti Aushadhe* (home remedies), *Gharcha Vaidya* (home doctor) and *Aajicha Batwa* (grandma’s wallet) are popular for collections of therapies that are either first-aid or therapies that can be easily administered by a nonpractitioner.

Compilations of various folk and traditional therapies are available. These compilations include therapies obtained by interviewing tribal people or therapies from available traditional literature [10,11,12,13,14,15,16,17,18,19]. The correct interpretation of available information and further research in the same direction is essential for the discovery of newer therapeutic molecules. A few plant extracts or isolated compounds have been tested systematically for the potential to neutralize snake venom, but there is a long way ahead for these molecules to have clinical utility.

## 2. Materials and Methods

In the current work, therapies for snake envenomation were gathered from traditional compilations, *Vanaushadhi Gunadarsha* Vol. 1–7 [20] and *Aushadhi Baad* Vol. 1–3 [9]. As these books garner therapies from historical times, the language is more traditional than the one used in the present day. These therapies are translated into English, and the plant species used therein are identified using internet search engines. Some vernacular plant names that were more traditional were not identified through internet sources and were identified from another book, *Gharguti Aushadhe* [21]. This book lists various traditionally used therapeutic plants along with their properties, method of use, dose and, in some cases, binomial names, as well. Validation of the binomial names of the plant species was performed by referring to the International Plant Name Index, available at https://ipni.org (accessed during 1–15 March 2022) and/or Plants of the World Online, available at https://powo.science.kew.org (accessed during 1–15 March 2022). Related species and changes in binomial names of the same species of the plant were identified from this exercise.

Individual plant species were then subjected to a modern literature search for parallel mention of snake venom remedies from other traditional or folk literature. Relevant pharmacological, therapeutic or molecular investigations of the plant species or their isolated chemicals against whole or components of snake venoms were searched and documented simultaneously. For this exercise, major scientific databases including Academia, CAB Abstracts, Chemical Abstracts Service, CiteSeerX, Cochrane Library, Crossref, EMBASE, Europe PMC, Index Copernicus, Indian Citation Index, J-Gate, MEDLINE, PubMed Central, ResearchGate and ScienceDirect were accessed in combination with Google Scholar search engine. This extensive search was carried out using binomial names of the plants coupled with the terms, “snake venom”, “cobra venom”, viper venom”, “snake poison”, “cobra poison”, “snake bite”, “cobra bite”, “krait bite”, “venomous bite” and “poisonous bite”. Wherever one plant was identified by multiple binomial names, all available names were included in the search. Manual screening of the bibliographies of the selected references was also performed to identify source information and other potential studies.

For any reference to qualify for inclusion in the current work, the following criteria were applied.

The article mentions the use of the whole plant, its part or extracted component as a therapy to treat snake envenomation and has ethnobotanical or folklore evidence.The article describes pharmacological research related to the neutralization of snake venom or its component by the specified plant, its part or extracted component.Multiple articles that cross-refer the same source of traditional use or pharmacological research without additional or unique particulars were not included.

This work mainly focuses on lesser investigated plants having potential for snake venom neutralization and discusses the rationality of the therapies mentioned. Plant-derived chemicals with potential pharmacological action have not been discussed to a greater extent.

## 3. The Traditional Therapies for Snake Envenomation

For the treatment of snake envenomation, a total of 51 therapies were identified of which 36 include single plant species, 13 include two to four plant species and 2 are large formulations including eight or more plant species. The therapies were listed, and the plant species were identified for further investigation in modern literature. Unlike modern medicine, therapies found in the traditional literature do not always have the exact dose or composition mentioned. As most of these were private journals held by the medical practitioners, they knew “how to” and “how much to” for these therapies. An attempt has been made to document all possible information regarding the therapies. The formulation technique, dose and direction for therapies along with identified plant species are listed in Table 1 and Table 2.

### 3.1. Large Formulations for Snake Envenomation

Few of the traditional Indian formulations comprised multiple ingredients and were sometimes intended to be used for a variety of ailments. It was either the dose of the formulation or its further treatment that decided what it would be used to cure. Similarly, a couple of formulations were noticed that were claimed for treatment of snake envenomation: *Nagachya Golya* (cobra pills) [9] (p. 171) and *Vyadhiharak Vatika* (curative pills) [9] (p. 61). *Nagachya Golya* is a formulation comprising 10 ingredients, of which 9 are plant-based. Formulation: Powdered Indian aconite 10 g, Mount Atlas daisy 10 g, long peppercorns 10 g, black peppercorns 5 g, nutmeg 10 g and mace 5 g should be combined with juice made from musk 5 g, ginger juice 20 g, holy basil leaf juice 30 g and betel leaf juice 10 g. The dough should be made, and small pills with a diameter of about 2–3 mm should be prepared [9] (p. 171). Identified plant species from the formulation are listed in Table 3. The information regarding the dose and administration of these pills was not found. The only nonplant ingredient in this formulation is musk, which is an aromatic substance obtained from the glandular secretions of male musk deer—*Moschus cupreus* (Moschidae). Musk is very precious, and finding genuine musk is hard in the present day as the musk deer is an endangered [22] animal species.

The second formulation, *Vyadhiharak Vatika*, is a generalized formulation used for various ailments. It is a 26-ingredient formulation including 21 plant species and 4 mineral substances. Formulation: aconite, borax, purified orpiment, long pepper, chebulic myrobalan, Indian gooseberry, Ceylon leadwort, sulphur, purging croton, Mount Atlas daisy, freshwater mangrove, castor, liquorice, babreng, sacred fig, Indian barberry, Indian aconite, henbane, mace, nutmeg, opium poppy, rock salt, pushkarmool, asafoetida and garlic should be soaked and levigated in false daisy juice for three days. Rosary-pea-sized pills should be prepared [9] (p. 61). To cure snakebite, one pill should be levigated with water and instilled into the eyes [9] (p. 61). Identified plant species from the formulation are listed in Table 3. Borax, orpiment, sulphur and rock salt are the mineral ingredients used in this formulation. The quantities of the ingredients to be taken are not mentioned in the literature and hence, as per convention, it should be regarded that all ingredients are taken in equal weight ratio. These formulations were prepared and stored by traditional practitioners for a longer time and used quickly when necessary.

### 3.2. Therapeutic Anomalies

Surprisingly, some of the mentioned therapies use an emetic and claim that the victim would vomit out snake venom, which is practically unjustifiable [9] (pp. 15, 124, 260), [20] (V. 1, p. 54) (V. 5, p. 357). Only orally ingested poisons can be thrown out via emesis to avoid further entry into the systemic circulation. Snakebite introduces venom directly into tissues and systemic circulation, which never can be thrown out via vomit. Though such therapies can be weeded out, they have been mentioned for documentation. In another atypical therapy, the snakebite victim should be made to sit on a platform made up of cow dung, and a stream of cold water should be poured on their body while administering the medication [20] (V. 3, p. 164). There appears to be no direct link between this physical component and the remaining plant-based therapy. Though these parts of therapies appear unjustified, some scientific research may be conducted before rendering the idea futile.

### 3.3. Toxicity of Plants

Many plants are known to have compounds with potent pharmacological action or severe toxicity. The dose of the isolated component or plant-based formulation decides toxic or therapeutic action. Many plants covered in Table 1, Table 2 and Table 3 have toxic potential and should be used with caution to achieve the desired effect. Castor—*Ricinus communis*—has a toxic water-soluble glycoprotein, ricin, that is considered a potential chemical weapon [23]. Ricin is present in the castor beans, and the above-listed therapies mention either the use of castor leaves juice or castor oil. It might be concluded that the toxicity of ricin would not be a problem in these formulations. Latex of apple of Sodom—*Calotropis procera*—is also known to have toxic proteins. However, the toxicity of these proteins is profound with oral administration [24]. Most methods listed above mention topical use of latex that might not be harmful. Aristolochic acid from *Aristolochia* sp. is reported as nephrotoxic, mutagenic and carcinogenic. It produces quick-progressing nephritis resulting in renal failure [25]. Cases of bristly luffa—*Luffa echinate*—toxicity have been reported upon oral consumption; however, after proper treatment, the patient did not have long-term ill effects [26]. Other plants such as purging croton—*Croton tiglium*—and Colocynth—*Citrullus colocynthis*—have toxicity reports [27,28]. While researching these plants, their margin of toxicity should be well considered while titrating the dose.

## 4. Snake Envenomation Prophylaxis

As alluring as it may sound, traditional literature also cites prophylactic remedies for snake envenomation. Being immune to snake envenomation might be fascinating for the snake handlers and farm workers. There are increasing incidences of snakebites among experienced snake handlers [29], and no prophylactic remedy is available yet. Having specific antibodies against snake venom seems the only viable hypothesis today as prophylactic. To achieve prophylaxis for snake envenomation, the agent should be well circulated through the systemic circulation, not be cytotoxic and should quickly inhibit systemic as well as local effects of venomous snakebite. Literature mentions *Azadirachta indica* A.Juss. (Meliaceae) or neem to be prophylactic against snake envenomation. If neem leaves are chewed daily in the morning, snake venom will not affect in case of a bite [20]. Another reference mentions “brushing the teeth daily with the stick of *Azadirachta indica* A.Juss. makes the body resistant to snake venom” [30]. However, no clinical or in vivo evidence is available to support the claim.

In Indian mythology, the concept of *Vishakanya* is noteworthy [31,32]. A *Vishakanya* is a lady who has venom running in her blood which she could infuse into someone’s body and kill. As the say goes, a small amount of snake venom is put on a stone, and the girl would lick it. The quantity of venom is increased gradually, and she would become a *Vishakanya*. It is hard to believe that venom would be retained in her body and can be used to affect someone else. However, it seems possible that with the sublingual introduction of sublethal doses of snake venom, antibodies were developed, rendering her immune to any future snake envenomation. It would be fair to think that the *Vishakanya* might handle snakes fearlessly and make them bite the target person. Modern literature evidence on *Vishakanya* is not available.

## 5. Snake Repellents

While searching for the prospective snake venom antidotes, a few remedies to repel snakes were also found. One of those is the entry deterrent described as, “To keep snakes away, a root of Indian fumitory should be kept at home” [9]. Indian fumitory—*Fumaria indica* Lam. (Papaveraceae)—is used in *Ayurvedic* medicine to treat ailments such as pain, diarrhoea and fever and has pharmacological evidence [33], but its use as snake repellent has not been mentioned earlier. Similarly, Asafoetida—*Ferula assa-foetida* L. (Apiaceae)—is also mentioned to repel snakes [34]. Apple of Sodom—*Calotropis procera* Aiton (Apocynaceae)—is also referred to as snake repellent. It is said that venomous snakes including cobras cannot withstand the smell of *Calotropis procera*, and thus snake charmers use it for controlling snakes [35]. One more remedy used to shrug away snakes from the hiding says, “Small pieces of the horn of Sambar deer should be burned. The snake (cobra) flees because it can’t stand the smoke” [9]. This is one of the few remedies that use agents of an animal origin. Sambar deer—*Rusa unicolor* (Cervidae)—is a threatened [36] animal found in the Indian subcontinent. Sambar stags have highly precious horns. Traditionally Sambar horns were used in the treatment of various diseases [37], but now their use has declined to almost nothing.

## 6. Routes of Administration

Traditionally, the routes of administration of medications were limited, and the injectable routes were not practised then. Oral, topical, nasal, ophthalmic, urethral, vaginal and rectal routes were in routine use in the earlier days [38]. Oral administration and topical application on the affected part were the most common routes used. During a snakebite, the venom is released into the tissues and has direct access to the systemic circulation of the victim. The bite site, bite intensity, amount of venom injected and type of venom decide the further course of spread, morbidity or mortality. A superficial bite in subcutaneous or muscular areas delays the spread of venom to other vital organs through systemic circulation and provides grace time to the victim for treatment. However, an intense bite with close access to a vein quickly deteriorates the situation and starts affecting the victim’s vital organs. In such cases, even short delay would result in death or permanent disabilities [39]. Dry bites have also been observed where venomous snakes might not release venoms during a bite in rare instances. There are no systemic symptoms during the dry bites, but local tissue inflammation is still observed. Management of snakebite is tricky due to all these factors, and accordingly selected agents are used. Various routes of administration practised in the earlier days do not seem practical in such acute situations where the spread of venom is faster and the availability of the drug is slower. Routes of administration described in the enlisted therapies for snake envenomation are discussed below.

### 6.1. Oral

Like most of the therapies, the oral route is the most common route of administration found in the listed therapies. Out of 51 therapies, 37 involve oral administration of formulations. Most of these formulations are either aqueous or are juices of plant parts. Most of the therapies mention the formulation of paste by trituration or levigation of the plant part. In modern pharmaceutical formulations, a penetration enhancer is added to improve or accelerate the absorption of the main drug. It is expected that the earlier multi-ingredient formulations had ingredients to augment the effect of the main drug and mask the taste or reduce any side effects. As it has been established in modern literature that ingredients such as black pepper, long pepper or ginger are used to improve the bioavailability of the principle content of the formulation [40,41], similar ingredients are used in the traditional formulations, as well. Additionally, some of the additives are proven to have pharmacological effects [42,43,44].

One particular therapy mentions the administration of colocynth root (*Citrullus colocynthis*) in *Paan/Vida*. Its most basic preparation is made with betel leaf combined with slacked lime, areca nut (*Areca catechu*) and *Kattha* (cold crystalized aqueous extractives of *Acacia catechu* heartwood). Consumption of *Paan* increases salivation, and probably some contents are absorbed by the buccal route. The intention behind the administration of colocynth in *Paan* might be faster absorption into the systemic circulation and maybe taste masking. The simple *Paan* preparation is mostly enriched with other ingredients such as fennel seeds (*Foeniculum vulgare*), clove buds (*Syzygium aromaticum*), cardamom seeds (*Elettaria cardamomum*), cinnamon (dried inner bark of *Cinnamomum verum*), tobacco (*Nicotiana tabacum*), rosary pea leaves (*Abrus precatorius*), dried coconut (*Cocos nucifera* kernel), *Dhana dal* (roasted and split coriander seeds), *Gulkand* (the sweet preserve of rose petals), menthol and many more. Though consumption of *Paan* is socially well accepted in India and a few other Asian countries, long-term consumption is associated with an increased risk of oral ailments including cancer [45,46,47,48]. In another therapy, oral administration of colocynth extract/juice is expected to cause emesis and purgative action. Probably to suppress emesis, in earlier therapy, it is directed to be consumed in *Paan*.

### 6.2. Topical

Effects on the local tissue near snakebite are manifested immediately after the bite. Local tissue effects are profound in cases of semipoisonous snakebites, as well. To tackle the local effects of snakebite or to neutralize the amount of venom that is yet to enter the systemic circulation, topical application on skin or affected tissue is an important route of administration. It seems to be the second-most practised route among the enlisted therapies. Application of formulation on the skin or bite area is mentioned in 17 therapies. Most of the formulations mentioned are aqueous or make use of juice from fresh plant parts. Commonly, pastes prepared by levigation of fresh or dried plant parts (in water) are used. Occasionally, grated fresh or processed plant parts are also applied directly to the wound. Application of formulations on the skin in a thick or thin layer, rubbing liquid and using a poultice were common practices in earlier days. Thick layers of formulation and poultices were applied on, surrounding or near bite sites. Local absorption of the medication might reduce inflammation, neutralize localized venom or prevent its further spread. These therapies seem rational and are the fastest to reach and neutralize venom before it reaches the systemic circulation causing further complications. A few therapies also involve rubbing the formulations away from the bite site. Quick absorption of the medication from the skin and probably into the systemic circulation might be the goal of such therapies. Another goal might be to protect vital organs in the abdominal region.

In a therapy known from Ceylon (Sri Lanka), the use of leeches is mentioned. These leeches are to be applied around the bite site along with the application of key lime juice [9] (p. 114). Leeches were used in traditional Indian medicine, and in the modern setup, their use is increasing, mainly for poorly healing wounds [49]. Leeches are known to suck blood from the area, and it is rational to think that localized venom would also be sucked out with it. This might prevent further entry of venom into systemic circulation but carries a side effect of excessive bleeding [50]. The type of venom would be of key importance. The use of this therapy in neurotoxic envenomation might still prove beneficial over cytotoxic envenomation where the coagulation cascade is hampered. Leeches also inject anticoagulants locally to facilitate drawing blood. In such cases, excessive bleeding might be fatal. In the older days when chances of death were much higher, such therapy still might have saved a few lives. Case reports of leech therapy in chronic snakebite wounds are available [51], but no direct evidence on the use of leeches to tackle snake envenomation was found. In contrast, some snake venom proteins have direct relevance to components injected by leeches [52]. Thus, in some instances, leech therapy might quickly worsen snakebite wounds locally.

### 6.3. Ocular

Following the most commonly used oral and topical routes, ocular administration comes in third place, with fewer mentions in various therapies. Instillation of aqueous formulations (drops) is most commonly found in the ocular therapies, where plant parts are levigated and diluted with water to form a suspension or a solution. In fewer therapies administration of *Anjana* (cream, ointment or paste for instillation in eyes) [38] is recommended. Plant parts are powdered, triturated or levigated with water or juice from the same or other plants to form *Anjana*.

In the case of snake envenomation, the local effect of instillation in the eyes does not seem very effective as a treatment. The medication should be absorbed into the systemic circulation for therapeutic effect. As the instilled medication travels through the lacrimal duct, into the nasal mucosa and pharynx, it absorbs into the systemic circulation. A small amount of drug also may absorb systemically through the conjunctival and other ocular blood vessels [53]. Some of the ingredients such as Indian soapberry or garlic, mentioned in the eye instillation therapies, might cause severe irritation to the eyes. This might be by design to increase the permeability of blood capillary, thereby facilitating faster absorption. Indian soapberry is mentioned as an antidote for various poisons and venoms, and it causes severe eye irritation. To tackle the irritation to the eyes, it is mentioned that clarified butter should be instilled later. Moreover, since large quantities cannot be administered through this route, the drug has to be highly potent, and the formulation should be concentrated.

### 6.4. Nasal

Nasal drug delivery for the treatment of snakebite is quite uncommon, and only a couple of therapies mention nasal delivery. Both solids and liquids can be administered through the nasal route. One of the therapies involves the instillation of a diluted aqueous paste into the nose while another mentions insufflation of a finely powdered mixture of ingredients. Insufflation would be possible if the snakebite victim is well awake and can quickly sniff the powder mixture. Passive administration by this route would be difficult. Though systemic drug delivery is possible with the nasal route, the drug has to be potent, as large volumes cannot be administered via this route [54]. As hepatic first-pass metabolism is avoided by nasal drug delivery [55], it seems favourable for some molecules to reach systemic circulation and neutralize venom components.

## 7. Contemporary Exploration of Plants as Antidotes for Snake Venom

The various snake envenomation treatments listed in Table 1 and Table 2 involve 66 plant species from 39 families. The highest number of species (five each), were from Apocynaceae and Asteraceae; four each from Euphorbiaceae; three each from Cucurbitaceae, Fabaceae, Piperaceae, Solanaceae and Zingiberaceae; two each from Malvaceae, Moraceae, Papaveraceae, Ranunculaceae and Rutaceae; and the rest were one from each family.

While identifying plant species and their binomial names based on the reference from traditional literature, sometimes more than one plant species was identified for one traditional name. A classic example is *Rui*, which directs to two closely connected species, *Calotropis procera* and *Calotropis gigantea*. In such cases, the modern literature search was conducted for both species. In addition, binomial names of some plants are revised occasionally by botanists, and the literature is available for both older and revised names. In such cases, the modern literature search was performed for all available names of the plant. Table 4 exhibits ethnobotanical and pharmacological evidence for each plant species as a therapy for snake envenomation.

## 8. Plant Parts

Many articles from modern literature give a fair idea about plant parts used as snake venom antidotes. These references are from both traditional mentions and pharmacological research. Different plant parts might hold specific chemicals that bring about desired pharmacological effect making them important for the particular action. Figure 1 depicts the frequency of various plant parts appearing as snake venom antidotes from modern literature. The highest number of mentions were found for extracted or isolated components followed by roots and leaves. Various remedies from the traditional literature mention utility of roots and leaves as useful plant parts. The remedies where no specific plant part was referred are also significant in number. Seeds, fruits and stem bark hold intermediate places in utility, and all other plant parts were sparsely used in the therapies.

Various traditional systems of medicine including *Ayurveda* have recited the importance of plant roots in medicine. Numerous bioactive compounds have been derived from the roots of various plant species [236]. Modified roots act as repositories of the components absorbed from the soil and synthesized within the plant, and many of them have medically important bioactives. Similarly, leaves are centres of biosynthesis of pharmacologically active components and utilize the constituents absorbed from the environment. All other plant parts have bioactives in relatively lower concentrations, and the same has been reflected in the current work.

## 9. Snake Species with Modern Evidence of Venom Neutralization

Table 4 depicts pharmacological evidence of snake venom neutralization by plants from modern literature. Various researchers have used in vivo, in vitro, and in silico methods or their combination to assess the neutralization of snake venoms by plants. The most references are available for *Naja naja* venom [64,74,94,96,108,173,203,212,226], followed by *Daboia russelii* venom [42,44,77,104,106,107,109,110,144,145,156,173]. Both these snake species are perfect representatives of their families Elapidae and Viperidae, respectively, and are the most occurring venomous snake species in the Indian subcontinent. Surprisingly, venoms of other two common snake species, *Bungarus caeruleus* [96,173] and *Echis carinatus* [75,78,104,173,226], have been studied less frequently. Only two more snake species from the Indian subcontinent, *Naja kaouthia* [104,144,167] and *Naja naja karachiensis* [60,61,62,128], have been found to have evidence for plant-based snake venom neutralization. All other venomous snake species are from tropical African or South American regions. Based on the mechanism of antagonism, the references were grouped into the following three categories.

### 9.1. Pharmacological Action

Snake venoms, being complex mixtures of toxins, enzymes and proteins, act on various tissues and systems simultaneously. Inhibition of venom lethality is the most important pharmacological outcome of any therapeutic agent for snake envenomation, and the highest number of references (25) was observed for this action. As snake venoms interfere with the coagulation cascade to a large extent, antihaemorrhagic (17) and anticoagulant (9) properties were subsequent. Immunomodulatory and hepatoprotective actions and competitive inhibition were a few more actions mentioned in the reports. Reduction in median lethal dose (LD_50_) of venom is the most studied method while studying pharmacological action followed by modulation of the coagulation cascade. Figure 2 depicts the number of appearances in references for plants that have pharmacological action against the venoms of respective snake species, and Table 5 enlists snake species wise references (label: Action).

### 9.2. Enzyme Inhibition

The most important peptides interfering with the normal physiological function of the body are the enzymes present in snake venoms. The impact of plant derivatives or extracts on individual enzymes has been widely studied by researchers while searching for a plant-based snake venom antidote. Prospective studies have indicated positive results of inhibition of various enzymes such as Phospholipase (5) and Phospholipase A2 (17), Fibrinolytic enzymes (9), Protease (7), Superoxide dismutase (5), Lipid peroxidase (5), Acetylcholinesterase (3), 5′-nucleotidase (2), Hyaluronidase (2), L-amino acid oxidase (2) and ATPase (1). Phospholipase A2 is a highly investigated enzyme responsible for an inflammatory response that increases the permeation of other venom components into the systemic circulation. Refer to Figure 2 and Table 5 for snake venom enzyme inhibition by plants (label: Enzyme X).

### 9.3. Toxin Inhibition

Toxins in snake venom impair critical physiological functions in the victim’s body. Though the majority of references were found on enzyme inhibition, a few references on toxin inhibition were found. *Naja naja* venom being most investigated, and the highest number of reports was on neutralization of neurotoxins (6). As *Daboia russelii* was the second-highest investigated species, following neurotoxins, and reports on myotoxins (5), cardiotoxins (2) and cytotoxin (1) were recorded. Refer to Figure 2 and Table 5 for snake venom toxin inhibition by plants (label: Toxin X).

## 10. Plant Species with Modern Evidence of Snake Venom Neutralization

Of the 66 plant species identified from traditional literature, 17 have modern evidence indicating the neutralization of various snake venom components. *Aristolochia indica* precedes with most references [17,69,74,86,88,97,98,99,100,101,102,103,104,105,106,107,108,109,110] in both traditional and modern literature. In *Marathi,* it is known as *Saapsund*. The first half of the name *Saap* means snake, and it occurs in many traditional references as a cure for snakebite. It shares the first position with *Emblica officinalis* [64,118,143,144,145] which is a medicinally important plant in India. Various references demonstrate its effectiveness as an antidote for snake venom. *Azadirachta indica* [17,64,69,74,151,164,165,166,167,168,169,170,171,172,173] and *Rauvolfia serpentina* [17,18,69,87,88,89,90,91,92,93,94,95,96] hold second place in pharmacological evidence. Figure 3 depicts the number of appearances in references for each plant species as a snake venom antidote (labels as mentioned in the earlier section).

## 11. Opportunities with a Few Additional Plant Species

Although there is a wealth of literature on plant-based therapies for snake envenomation, some plant species are notably absent. In the modern literature, 18 of the 66 plant species discussed in this work have no evidence as an antidote for snake envenomation. These plants have been investigated for other pharmacological activities but not for snake venom. This opens up research opportunities and indicates that most of these plant species are medicinally important. Table 6 lists these plant species and their families. These plants should be investigated for potential small molecules acting as snake venom antidote.

## 12. The Way Forward

All plant-based therapies mentioned in the traditional or modern literature lack viable clinical evidence of being significantly effective against snake envenomation. Chemical components derived from whole plants or their parts have only been tried as a therapy against snake envenomation in either in vivo animal models, in vitro models or a few in silico models. The experimental design of these studies does not mimic the actual post-snakebite situation [237]. Some of the studies mentioned in vitro mixing of plant extracts with snake venom before injection in animals, while other studies mention the administration of test agent immediately after envenomation. For preliminary screening, such an experimental setup might work, but eventually, models mimicking actual clinical situations should be developed. The majority of the agents found effective in the initial studies might be rendered ineffective in the actual situations. Effective experimental models and in-depth studies are needed for an agent to even qualify for a clinical study. Apart from the herbal front, biotechnological approaches are also being explored. Monoclonal antibodies or other proteins are being tested as possible antidotes. An article mentions the use of a ribosomal peptide made up of 11 amino acids that can neutralize rattlesnake venom [238]. However, once again, this warrants premixing of proteins with venom before injecting into the animals.

Many plant species from Table 4 and the entire list in Table 6 have no pharmacological evidence of being useful as snake venom antidotes. Various traditional therapies describe combinations of multiple plant products in which one or more plant derivatives might have a principal active constituent(s) while others might be adjuvants such as bioavailability enhancers. Similar to modern medicine, *Ayurveda* has mentioned the concept of *Yogavahi* in its traditional literature, which is a bioavailability enhancement. *Glycyrhhiza glabra* L., *Piper longum* L. and *Zingiber officinale* Rosc. have been used classically as bioavailability enhancers [239]. Some plants might also be useful in reducing toxicity or other side effects imparted by the principal constituent plant. A detailed study based on plant constituents and their known roles from modern literature would help classify the plants and would create a platform for further experimental investigations.

In the rural areas of tropical countries, access to sophisticated medical treatment is scanty, or medical help is so far, that a snakebite victim succumbs on the way to medical facilities. Administration of snake venom antiserum is another challenge, and the dose titration is usually based on the symptoms presented by the victim [237] and tolerance to the SVA during therapy. Immediate hypersensitivity reactions associated with equine serum proteins make therapy more complicated [240]. World Health Organization (WHO) has published clear guidelines for standardization and refining of snake venom antisera [241], which have reduced the reactions. The availability of intensive care units and expert clinicians is primary in the treatment making the use of SVA. Though the availability of SVA has improved over the years, having it handy at the right time in the right place is crucial. There still is a big limitation to this SVA available in India as it is limited only to the “Big Four” snake species. It is effective only against venoms of *Naja naja*, *Bungarus caeruleus*, *Daboia russelii* and *Echis carinatus.* This polyvalent snake venom antiserum has very limited or no use in bites of other venomous snake species, and there is a dire need to develop antisera for the venoms of other clinically important snake species, as well [242]. The coastline of the Indian and Pacific Oceans faces another peculiar threat of sea snake envenomation. Warm currents of the Indian Ocean have many venomous sea snake species [243]. Accidental exposure of fishermen to sea snakes makes them vulnerable to envenomation. Sea snake venom is more toxic than that of terrestrial snakes, including cobras, but a sufficient quantity of venom is seldom injected during the bite [244]. Thus, overall mortality of sea snake envenomation is lower. Myoglobinuria is a common symptom that differentiates sea snakes from land snakes [244]. Though antiserum is available for sea snakes internationally and in most South-East Asian countries, its availability in India is uncommon. 

The mental trauma associated with snakebite also plays a vital role in worsening clinical symptoms. The more anxious the patient is, the more are the chances of exacerbation of clinical symptoms. The stress associated with snakebite brings psychoneuroendocrinological effects into action [245]. Even today, snakebite is believed to be the act of evil or the wrath of the goddess in some rural cultures, and the victims are taken to local healers who are believed to have magical powers. Some social groups are working on superstition eradication and educating people to seek immediate medical help postsnakebite as the delay can result in severe morbidity or mortality. During field work, one such event was witnessed in the *Konkan* region of *Maharashtra*, where a farm worker snakebite victim was taken to a healer instead of seeking medical help. Deprived of survival hopes, the victim was suffering from tachycardia and palpitations. The victim’s coworkers killed the snake after a bite and carried it with them. In absence of the *Vaidu* (traditional healer), another person approached them and carefully looked at the situation. He chanted some *mantras*, prepared lemonade and asked the victim to sit across and drink the lemonade. He continued chanting and made the victim drink another glass of lemonade a few minutes later. The victim started feeling better, and the tachycardia and palpitations were relieved. After everyone was relieved of the death threat, the person revealed his identity as being a person from a superstition eradication group. He explained how the semipoisonous snake caused local tissue inflammation, was never a threat to the life and the tachycardia and palpitations were due to severe anxiety. The victim was then taken to a healthcare centre and treated further. The local people would never have taken the victim to the healthcare centre if it was not done this way.

Snake envenomation is a real neglected tropical disease. Its incidence is not as common as diseases such as diabetes or hypertension, as these lifestyle disorders do not kill a person overnight and have a lot of potential for the sale of medications to manage the disease over a long time. Snake envenomation, on the other hand, is an acute situation for which the time frame for its management is very short, and consumption of drugs is limited. Naturally, due to high return potential, research investment is much higher for lifestyle disorders in comparison to that of snake envenomation. A small molecule with a high margin of safety, which can be self-administered by a victim or a primary healthcare worker, would be a great life-saver in case of snake envenomation. Natural resources such as plants and traditional literature might have highly effective hidden gems. A lot of research would be necessary before any such molecule becomes clinically useful.

## Figures and Tables

**Figure 1 tropicalmed-07-00108-f001:**
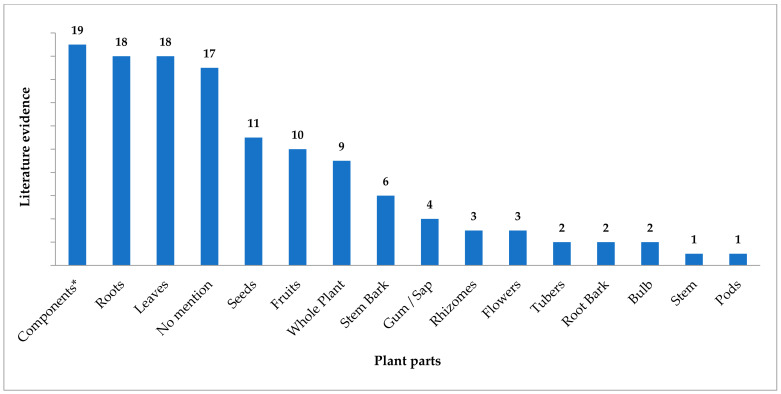
Plant parts as an antidote for snake envenomation from modern literature. * Extracted components or chemicals.

**Figure 2 tropicalmed-07-00108-f002:**
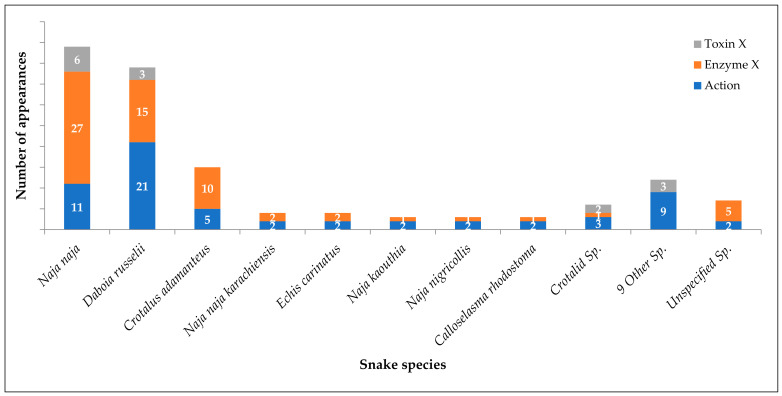
Modern literature evidence on snake species-specific inhibition of venom and its components.

**Figure 3 tropicalmed-07-00108-f003:**
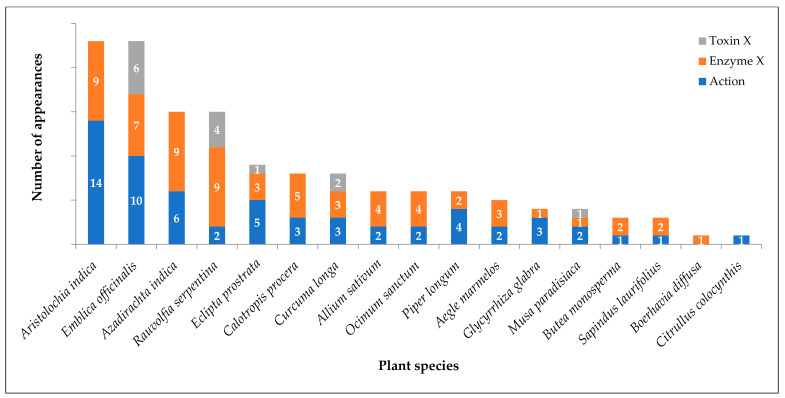
Modern literature evidence on plant species that inhibit snake venom and its components.

**Table 1 tropicalmed-07-00108-t001:** Snake-bite therapies comprising single plant species.

Formulation/Dose/Direction	Plant Common Name	Botanical Name (Family)
To treat cobra, Russell’s viper or saw-scaled viper envenomation, juice of Indian snakeroot should be orally administered [20] (V. 3, p. 417).	Indian snakeroot	*Rauvolfia serpentina* (L.) Benth. ex Kurz (Apocynaceae)
Four parts of castor leaf juice diluted with one part of water should be taken orally, and a paste of the leaves should be applied to the bite area. The individual will vomit venom [20] (V. 1, p. 54).	Castor	*Ricinus communis* L. (Euphorbiaceae)
Prickly pear root levigated in cow milk should be administered twice daily. Eating spicy foods should be avoided [20] (V. 4, p. 258).	Prickly pear	*Opuntia elatior* Mill. (Cactaceae)
Sacred tree root levigated with water should be orally administered and a thick paste should be applied topically to the bite area [20] (V. 4, p. 273).	Sacred tree	*Butea monosperma* (Lam.) Taub. (Fabaceae)
To treat krait venom: 5–10 mL of Portia bark juice should be taken orally [20] (V. 4, p. 279).	Portia tree	*Thespesia populnea* (L.) Sol. ex Corrêa (Malvaceae)
To treat cobra and saw-scaled viper venom: The juice of crowded-flower jasmine leaves should be administered orally as per tolerance [20] (V. 5, p. 381).	Crowded-flower jasmine	*Jasminum coarctatum* Roxb. (Oleaceae)
To treat snake or Russell’s viper venom: Levigated paste of white Nerium root should be applied topically on the bite area, or juice of leaves should be administered orally. In case of drowsiness due to this medication, clarified butter should be administered [20] (V. 2, p. 69).	Oleander or Nerium	*Nerium oleander* L. (Apocynaceae)
To treat krait envenomation: Levigated paste of spiny gourd tubers in honey should be instilled into the eyes, or juice should be administered orally [20] (V. 2, p. 74).	Spiny gourd	*Momordica dioica* Roxb. ex Willd. (Cucurbitaceae)
Levigated paste of spiny gourd tubers in water should be administered orally and applied topically to the stung area [20] (V. 2, p. 74).	Spiny gourd	*Momordica dioica* Roxb. ex Willd. (Cucurbitaceae)
Apple of Sodom leaves should be crushed with its sticky sap and formed into pills. These pills should be administered orally at regular intervals, or a levigated root paste should be administered orally [20] (V. 5, p. 360).	Apple of Sodom	*Calotropis procera* Aiton (Apocynaceae)
Levigated paste of sacred tree root should be administered orally or applied topically [20] (V. 4, p. 273).	Sacred tree	*Butea monosperma* (Lam.) Taub. (Fabaceae)
Levigated paste of creeping launaea root should be administered orally [20] (V. 4, p. 278).	Creeping launaea	*Launaea procumbens* L. (Asteraceae)
Colocynth root should be consumed in *Paan/Vida* (a preparation of betel leaf and areca nut made with slacked lime) [20] (V. 1, p. 38).	Colocynth	*Citrullus colocynthis* L. (Cucurbitaceae)
Levigated paste of conkerberry root in water should be administered orally [20] (V. 2, p. 75).	Conkerberry	*Carissa congesta* Wight. (Apocynaceae)
Filtered aqueous soapberry solution should be instilled in the eyes. In the case of severe envenomation, soapberry water should be administered orally so that the venom is vomited out. To avoid postinstillation irritation of the eyes, white butter or clarified butter should be applied [20] (V. 5, p. 357).	Indian soapberry	*Sapindus mukorossi* Gaertn. (Sapindaceae)
Snakebite test: A person not recognising the taste of neem leaves, salt or chilli peppers when given orally indicates snake envenomation. To treat envenomation, neem leaves should be chewed, or leaf or bark juice should be administered orally [20] (V. 2, p. 63).	Neem	*Azadirachta indica* A.Juss. (Meliaceae)
In the case of skin lumps caused by saw-scaled viper venom: Warmed bitter cumin leaves should be applied topically, or its juice should be rubbed on the affected area [20] (V. 2, p. 61).	Bitter cumin	*Centratherum anthelminticum* (L.) Kuntze. (Asteraceae)
For cobra venom: Cluster fig bark paste diluted with the mixture of its juice and milk should be administered orally [9] (p. 89).	Cluster fig	*Ficus racemosa* L. (Moraceae)
In the event of a snakebite: Jaggery and sesame seeds should be crushed in cow’s milk and consumed orally [9] (p. 16).	Sesame seeds	*Sesamum indicum* L. (Pedaliaceae)
Crushed paste of prickly pear leaves should be applied topically [9] (p. 87).	Prickly pear	*Opuntia elatior* Mill. (Cactaceae)
Punarnava root levigated with water should be administered [9] (p. 87).	Punarnava	*Boerhavia diffusa* L. Nom. Cons. (Nyctaginaceae)
Powdered Punarnava roots should be administered orally with water [9] (p. 87).	Punarnava	*Boerhavia diffusa* L. Nom. Cons. (Nyctaginaceae)
Crushed coffeeweed root paste should be instilled in the eyes [9] (p. 99).	Coffeeweed	*Cassia occidentalis* (L.) Rose. (Caesalpiniaceae)
The root of the apple of Sodom levigated with water should be instilled into the nose and eyes [9] (p. 99).	Apple of Sodom	*Calotropis procera* Aiton (Apocynaceae)
Rosary peas levigated with water should be administered orally [9] (p. 99).	Rosary pea	*Abrus precatorius* L. (Fabaceae)
An aqueous solution of Indian soapberry should be administered orally until the patient vomits a couple of times. An aqueous paste made from Indian soapberry should be instilled into the eyes. Levigated paste of Indian soapberries should be applied topically to the stung area [9] (p. 15).	Indian soapberry	*Sapindus mukorossi* Gaertn. (Sapindaceae)
Powdered potato yam stem bark should be administered with water [9] (p. 19).	Potato yam	*Dioscorea bulbifera* L. (Dioscoreaceae)
A reference from Ceylon (Sri Lanka): The key lime juice was applied to the stung area, and leeches were placed around it. On the third day, a laxative was administered, and the person was cured [9] (p. 114).	Key lime	*Citrus aurantifolia* (Christm.) Swingle (Rutaceae)
One or one and a half bristly luffa fruits should be crushed in water, decanted and administered orally. Clarified butter should be administered after vomiting occurs [9] (p. 124).	Bristly luffa	*Luffa echinata* Roxb. (Cucurbitaceae)
To treat Russel’s viper bite: Apply castor oil topically on the bite [9] (p. 142).	Castor	*Ricinus communis* L. (Euphorbiaceae)
To treat krait envenomation: Oral administration of fire-flame bush juice is recommended [9] (p. 256).	Fire-flame bush	*Woodfordia fruticosa* (L.) Kurz (Lythraceae)
The juice prepared from a minimum of 12–18 g of colocynth or its water extract should be taken orally. Venom will be expelled out in the form of vomit or faeces [9] (p. 260).	Colocynth	*Citrullus colocynthis* L. (Cucurbitaceae)
The root of Indian night shade levigated with water should be administered orally and applied topically on the stung area [9] (p. 274).	Indian night shade	*Solanum indicum* L. (Solanaceae)
Bristly luffa fruit extract in 40 g of cold water should be taken orally twice a day [9] (p. 297).	Bristly luffa	*Luffa echinata* Roxb. (Cucurbitaceae)
Levigated paste of devil’s goad roots should be administered orally [9] (p. 297).	Devil’s goad	*Croton roxburghii* Balakr. (Euphorbiaceae)
Steamed grated unripe papaya should be applied topically for 2–4 days around the bite area [9] (p. 297).	Papaya	*Carica papaya* L. (Caricaceae)

**Table 2 tropicalmed-07-00108-t002:** Snake-bite therapies comprising 2–4 plant species.

Formulation/Dose/Direction	Plant Common Name	Botanical Name (Family)
Levigated paste of Ceylon Leadwort roots, chilla roots and *Kale Vel* bulb should be administered thrice orally at intervals. The snakebite victim should be made to sit on a platform made up of cow dung while a stream of cold water should be poured on their body. The effects of the venom would wear off in about 6 h. In case of adverse events, clarified butter can be orally administered [20] (V. 3, p. 164).	Ceylon leadwort	*Plumbago zeylanica* L. (Plumbaginaceae)
	Chilla	*Casearia graveolens* Dalz. (Salicaceae)
	*Kale Vel*	Unidentified
Freshwater mangrove fruits levigated with garlic juice should be instilled into the eyes [20] (V. 6, p. 403).	Freshwater mangrove	*Barringtonia acutangula* (L.) Gaertn. (Lecythidaceae)
	Garlic	*Allium sativum* L. (Amaryllidaceae)
A pill should be made from finely powdered purging croton seed kernels that have been coated with key lime fruit juice 12 times. This pill levigated with human saliva should be instilled into the eyes [20] (V. 3, p. 175).	Purging croton	*Croton tiglium* L. (Euphorbiaceae)
	Key lime	*Citrus aurantifolia* (Christm.) Swingle (Rutaceae)
A diluted paste of Indian birthwort made with soapberries or white abrus should be taken orally at regular intervals [20] (V. 6, p. 418).	Indian birthwort	*Aristolochia indica* L. (Aristolochiaceae)
	Indian soapberry	*Sapindus mukorossi* Gaertn. (Sapindaceae)
	White abrus	*Abrus precatorius* L. (Fabaceae)
To treat Russell’s viper envenomation: A thick paste of devil’s goad roots, soapberries, and bitter cumin made with veld grape juice should be diluted with cow urine and applied topically at the bite site [20] (V. 3, p. 155).	Veld grape	*Cissus quadrangularis* L. (Vitaceae)
	Devil’s goad	*Croton roxburghii* Balakr. (Euphorbiaceae)
	Indian soapberry	*Sapindus mukorossi* Gaertn. (Sapindaceae)
	Bitter cumin	*Centratherum anthelminticum* (L.) Kuntze. (Asteraceae)
To treat Russell’s viper and krait venom: Paste formed by levigating devil’s goad roots and dry ginger should be administered orally. It will act as purgative and emetic. In case of skin lumps due to envenomation, the levigated paste of devil’s goad roots should be applied topically [20] (V. 3, p. 152).	Devil’s goad	*Croton roxburghii* Balakr. (Euphorbiaceae)
	Dry ginger	*Zingiber officinale* Roscoe (Zingiberaceae)
A finely powdered mixture of dried white Nerium flowers, tobacco and cardamom should be insufflated [20] (V. 2, p. 69).	Nerium	*Nerium oleander* L. (Apocynaceae)
	Tobacco	*Nicotiana tabacum* L. (Solanaceae)
	Cardamom	*Elettaria cardamomum* L. Maton (Zingiberaceae)
To treat Russell’s viper venom: Levigated paste of devil’s goad roots in the juice of three parts hog plum bark, two parts grey downy balsam bark and one part frangipani bark should be administered orally as tolerated [20] (V. 1, p. 16).	Devil’s goad	*Croton roxburghii* Balakr. (Euphorbiaceae)
	Hog plum	*Spondias pinnata* (L.f.) Kurz. (Anacardiaceae)
	Grey downy balsam	*Garuga pinnata* Roxb. (Burseraceae)
	Frangipani	*Plumeria acutifolia* Poir. (Apocynaceae)
Jaggery and sesame seeds should be crushed in the sticky sap of apple of Sodom and consumed orally [9] (p. 95).	Sesame seeds	*Sesamum indicum* L. (Pedaliaceae)
	Apple of Sodom	*Calotropis procera* Aiton (Apocynaceae)
Powdered East Indian walnut seeds mixed with prickly pear sticky sap should be applied to the stung area [9] (p. 156).	East Indian walnut	*Albizia saman* (Jacq.) F.Muell. (Fabaceae)
	Prickly pear	*Opuntia elatior* Mill. (Cactaceae)
Equal portions of salted dried roselle, powdered limestone and powdered turmeric should be levigated in cow urine and applied topically on the bite area for 3–7 days [9] (p. 156).	Roselle	*Hibiscus sabdariffa* L. (Malvaceae)
	Turmeric	*Curcuma longa* L. (Zingiberaceae)
A person fainted as a result of snake envenomation. An ascetic rubbed holy basil juice around the victim’s navel, chest, and forehead and kept a cloth ball dipped in the holy basil juice until the victim woke up. The banana stem water was then administered orally [9] (p. 196).	Holy basil/Tulasi	*Ocimum tenuiflorum* L. (Lamiaceae)
	Banana	*Musa acuminata* Colla (Musaceae)
An amount of 30 g of Bengal quince bark or leaves juice, 30 g of Frangipani bark juice, and 3 pills of Sanjeevini should be administered together [9] (p. 297).	Bengal quince	*Aegle marmelos* (L.) Corr. (Rutaceae)
	Frangipani	*Plumeria acutifolia* Poir. (Apocynaceae)
	Sanjeevini	*Selaginella bryopteris* (L.) Baker (Selaginellaceae)

**Table 3 tropicalmed-07-00108-t003:** Plant species used in *Nagachya Golya* and *Vyadhiharak Vatika*.

Common Name	Botanical Name (Family)
Aconite	*Aconitum heterophyllum* Wall. (Ranunculaceae)
Asafoetida	*Ferula assa-foetida* L. (Apiaceae)
Babreng	*Embelia ribes* Burm.f. (Myrsinaceae)
Betle leaf	*Piper betle* L. (Piperaceae)
Black pepper	*Piper nigrum* L. (Piperaceae)
Castor	*Ricinus communis* L. (Euphorbiaceae)
Ceylon leadwort	*Plumbago zeylanica* L. (Plumbaginaceae)
Chebulic myrobalan	*Terminalia chebula* Retz. (Combretaceae)
False daisy/Bhringraj	*Eclipta prostrata* L. (Asteraceae)
Freshwater mangrove	*Barringtonia acutangula* (L.) Gaertn. (Lecythidaceae)
Garlic	*Allium sativum* L. (Amaryllidaceae)
Ginger	*Zingiber officinale* Roscoe (Zingiberaceae)
Henbane	*Hyoscyamus niger* L. (Solanaceae)
Holy basil/Tulasi	*Ocimum tenuiflorum* L. (Lamiaceae)
Indian aconite	*Aconitum ferox* Wall. ex Ser. (Ranunculaceae)
Indian barberry	*Berberis aristata* DC. (Berberidaceae)
Indian gooseberry	*Phyllanthus emblica* L. (Euphorbiaceae)
Liquorice	*Glycyrrhiza glabra* L. (Fabaceae)
Long pepper	*Piper longum* L. (Piperaceae)
Mount Atlas daisy	*Anacyclus pyrethrum* (L.) Link. (Asteraceae)
Nutmeg, Mace	*Myristica fragrans* Houtt. (Myristicaceae)
Opium poppy	*Papaver somniferum* L. (Papaveraceae)
Purging croton	*Croton tiglium* L. (Euphorbiaceae)
Pushkarmool	*Inula racemosa* Hook.f. (Asteraceae)
Sacred fig	*Ficus religiosa* L. (Moraceae)

**Table 4 tropicalmed-07-00108-t004:** Ethnobotanical and pharmacological evidence of plant species used in the treatment of snake envenomation in the modern literature.

Family	Botanical Name	Ethnobotanical Evidence (Plant Part) ^$^	Pharmacological Evidence
Amaryllidaceae			
	*Allium sativum* L.	Bulbs [18,56,57,58,59]	Ethanolic, methanolic and aqueous extracts of garlic (*Allium sativum*) when tested against *Naja naja karachiensis* venom by in vitro (hen’s egg yolk mixture) and in vivo (male rabbits) methods; Phospholipase A2 and snake venom proteins were neutralized, and snake venom proteins failed to bind to their potential targets due to hindrance by secondary metabolites of garlic [60,61,62]. Orally administered dose of garlic juice to rats for 10 days can be used as a prophylactic tool against cobra venom without significant side effects on gastric and hepatic tissues [63]. Garlic bulbs neutralized coagulant, fibrinolytic and phospholipase activity in vitro from *Naja naja* venom [64].
Anacardiaceae			
	*Spondias pinnata* (L.f.) Kurz. *Spondias mangifera* Willd.	Leaves, Fruits [65] Plant * [12]	None
Apiaceae			
	*Ferula assa-foetida* L.	Exudates [66]	None
Apocynaceae			
	*Calotropis procera* Aiton *Calotropis gigantia* L. Dryand	Whole plant [67] Roots [18] Root bark [68] Leaves [69,70,71] Flowers [72] Dried latex [73]	Organic solvent extracts of *Calotropis* inhibited various enzymes from *Naja naja* venom when tested in vitro [74]. Fractionated methanolic extract of *Calotropis procera* neutralized the activity of 5′-nucleotidase from *Echis carinatus* venom by 85% when compared with snake venom antiserum [75]. *Calotropis* inhibited proteases and phospholipase A2 found in snake venoms when tested in vitro [76]. The methanolic root extract of *Calotropis* has potent inhibitors of phospholipase A2 from *Daboia russelii* venom [77]. Methanolic extract of *Calotropis procera* exhibited some degree of protection against *Bitis arietans*, *Echis ocellatus* and *Naja nigricollis* snake venoms in albino rats [78].
	*Carissa congesta* Wight. *Carissa carandas* L.	Leaves [79,80] Plant * [81]	None
	*Nerium oleander* L.	Whole plant [69] Roots [82,83] Leaves [84,85] Seeds [86]	None
	*Plumeria acutifolia* Poir. *Plumeria lutea* Ruiz. *Plumeria alba* L. *Plumeria rubra* L.	Fruits [18]	None
	*Rauvolfia serpentina* (L.) Benth. ex Kurz	Whole plant [69] Roots [18,87,88,89,90] Roots, Leaves [91] Plant * [92,93]	The alkaloid fraction of *Rauvolfia serpentina* leaves neutralized *Crotalus adamanteus* snake venom by reducing venom lethality, Superoxide dismutase activity and lipid peroxidation [17]. Aqueous extract of roots of *Rauvolfia serpentina* neutralized enzymes such as Acetylcholinesterase, ATPase and Protease and lethality of *Naja naja* snake venom when conducted by in vivo and in vitro methods [94]. *Rauvolfia serpentina* contains molecules that can potentially inhibit various cobra venom enzymes such as Acetylcholinesterase, L-aminoacid oxidase, Phospholipase A2 and Serine protease when tested in silico via molecular docking techniques [95]. Neurotoxins from *Bungarus caeruleus*, *Dendroaspis polylepis polylepis*, *Naja naja* and *Oxyuranus microlepidotus* venoms depicted a high affinity for binding with phytochemical compounds from *Rauvolfia serpentina* when tested in silico via molecular docking studies [96].
Aristolochiaceae			
	*Aristolochia indica* L.	Whole plant [69,97,98,99,100] Roots [17,86,88,101,102,103,104,105]	Aqueous ethanolic extract of *Aristolochia indica* inhibited various enzymes from *Naja naja* venom when tested in vitro [74]. The alkaloid fraction of *Aristolochia indica* leaves neutralized *Crotalus adamanteus* snake venom by reducing venom lethality, Superoxide dismutase activity and lipid peroxidation [17]. Methanolic extract of *Aristolochia indica* neutralized lethal action of *Daboia russelii*, *Echis carinatus*, *Naja kaouthia* and *Ophiophagus hannah* venoms in male albino mice; haemorrhagic action of *Daboia russelii* and *Echis carinatus* venom in male albino mice; coagulant activity of *Daboia russelii* venom in vitro; defibrinogenating activity of *Daboia russelii* venom in male albino mice; Phospholipase A2 activity of *Daboia russelii* venom in vitro [104]. An aqueous root extract of *Aristolochia indica* elongated the duration of survival of animals after the application of Russell’s viper venom [106]. Methanolic extract of *Aristolochia indica* (whole plant) completely neutralized the lethality of *Daboia russelli* venom, and oedema, haemorrhagic, coagulant, fibrinolytic and phospholipase activities were reversed. *Aristolochia indica* plant extract has a potent activity to neutralize Russell’s viper venom [107]. Aristolochic acid from *Aristolochia indica* inhibited the in vitro activity of purified hyaluronidase from *Naja naja* venom as well as the overall venom hyaluronidase activity in a dose-dependent manner [108]. Mice injected with whole *N. naja* venom preincubated with aristolochic acid had a two-fold increase in survival time when compared to mice injected with venom alone. Aristolochic acid injection 10 min after whole venom injection resulted in a more moderate increase in survival time [108]. Aristolochic acid inhibited Phospholipase A2 from *Vipera russelii* venom in a noncompetitive manner [109]. A major constituent of *A. indica*, aristolochic acid, inhibited Russell’s viper venom L-amino acid oxidase enzyme activity by interacting with DNA [110].
Asteraceae			
	*Centratherum anthelminticum* (L.) Kuntze.	Seeds [111]	None
	*Eclipta prostrata* L.	Plant * [69,112]	The alkaloid fraction of *Eclipta prostrata* leaves neutralized *Crotalus adamanteus* snake venom by reducing venom lethality, superoxide dismutase activity and lipid peroxidation [17]. Butanolic extract of *Eclipta prostrata* partially inhibited the haemorrhagic activity of *Calloselasma rhodostoma* (Malayan pit viper) venom [113]. The lethal activity of *Crotalus durissus terrficus* (South American rattlesnake) venom was neutralized by ethanolic extracts of *Eclipta prostrata* (aerial parts) in mice when mixed in vitro before injection intraperitoneally [114]. *Eclipta prostrata* acts as antimyotoxic and antihaemorrhagic against crotalid venoms [115]. *Eclipta prostrata* ethyl acetate extract inhibited the proteolytic and haemorrhagic activity of *Calloselasma rhodostoma* (Malayan pit viper) venom [116].
Berberidaceae			
	*Berberis aristata* DC.	Berberine [117] Bulbs ^ [118]	None
Caesalpiniaceae			
	*Cassia occidentalis* (L.) Rose.	Roots [69]	None
Caricaceae			
	*Carica papaya* L.	Papain [69,119] Leaves [120] Plant * [121]	None
Combretaceae			
	*Terminalia chebula* Retz.	Fruits ^ [118] Fruits ~ [122,123]	None
Cucurbitaceae			
	*Citrullus colocynthis* L.	Fruits [124] Roots, Fruits [125] Roots, Fruits, Seeds [126,127]	Methanolic extract of the whole plant of *Citrullus colocynthis* neutralized haemorrhage induced by *Naja naja karachiensis* snake venom [128].
	*Momordica dioica* Roxb. ex Willd.	Roots [18] Roots/Tubers [129] Roots, Fruits [130] Fruits [131,132] Plant * [133,134]	None
Dioscoreaceae			
	*Dioscorea bulbifera* L.	Sap [135]	None
Euphorbiaceae			
	*Croton roxburghii* Balakr.	Stem bark [136] Plant * [12,137,138,139]	None
	*Croton tiglium* L.	Roots, Stem Bark, Seeds [140,141] Plant * [142]	None
	*Phyllanthus emblica* L. *Emblica officinalis* Gaertn.	Fruits ^ [118] Plant * [143]	Fruits of Indian gooseberry when tested in vitro for *Naja naja* venom-neutralizing capacity by evaluating coagulant activity, fibrinolytic activity and phospholipase activity were found effective [64]. Methanolic root extract of *Emblica officinalis* significantly neutralized lethal activities of *Naja kaouthia* and *Vipera russelii* venoms and reversed *V. russelii* venom-induced coagulant, defibrinogenating, haemorrhage and inflammatory activities [144]. A phthalate compound from *Emblica officinalis* root extract neutralized cardiotoxic, defibrinogenating, haemorrhagic, neurotoxic, myotoxic, proinflammatory and PLA2 activities induced by viper and cobra venoms [145].
Fabaceae			
	*Abrus precatorius* L.	Roots [17,69,146,147,148,149,150,151]	None
	*Butea monosperma* (Lam.) Taub.	Roots [152,153,154] Stem bark [155]	Organic solvent extracts of *Butea monosperma* inhibited various enzymes from *Naja naja* venom when tested in vitro [74]. Ethanolic extract of *Butea monosperma* inhibited *Vipera russelii* venom hyaluronidase in a dose-dependent manner, and thus, venom-induced haemorrhage was reduced significantly [156].
	*Glycyrrhiza glabra* L.	Roots [157]	Glycyrrhizin, from *Glycyrrhiza glabra* prevents *Bothrops jararaca* venom-induced changes in haemostasis both in vitro and in vivo [158]. Flavonoids from *Glycyrrhiza glabra* inhibit the activity of cytosolic and secreted phospholipase A2 in a dose-dependent manner suggesting anti-inflammatory and immunomodulatory actions when tested via in silico methods [159].
Lamiaceae			
	*Ocimum tenuiflorum* L. *Ocimum sanctum* L.	Flowers ^ [118] Plant * [160,161,162]	Tulsi leaves were tested in vitro for *Naja naja* venom neutralizing capacity by evaluating coagulant activity, fibrinolytic activity and phospholipase activity and were found effective [64]. The alkaloid fraction of *Ocimum sanctum* leaves neutralized *Crotalus adamanteus* snake venom by reducing venom lethality, superoxide dismutase activity and lipid peroxidation [17].
Lecythidaceae			
	*Barringtonia acutangula* (L.) Gaertn.	Seeds ~ [163]	None
Lythraceae			
	*Woodfordia fruticosa* (L.) Kurz	Whole plant [69] Leaves [18]	None
Meliaceae			
	*Azadirachta indica* A.Juss.	Whole plant [164] Leaves [69,151,165] Stem bark, Gum, Leaves, Seeds [166]	Neem bark was tested in vitro for *Naja naja* venom neutralizing capacity by evaluating coagulant activity, fibrinolytic activity and phospholipase activity and was found effective [64]. Organic solvent extracts of *Azadirachta indica* inhibited various enzymes from *Naja naja* venom when tested in vitro [74]. The terpenoid fraction of *Azadirachta indica* leaves neutralized *Crotalus adamanteus* snake venom by reducing venom lethality, superoxide dismutase activity and lipid peroxidation [17]. An isolated compound—AIPLAI—from methanolic leaf extract of *Azadirachta indica* inhibited phospholipase A2 from cobra and Russell’s viper venoms in a dose-dependent manner. AIPLAI inhibited purified phospholipase A2 from *Naja kaouthia* venom in a noncompetitive manner [167]. Fractionated leaf extracts of *Azadirachta indica* presented significant hepatoprotection against the *Naja nigricollis* venom-induced toxicity in albino rats [168]. Isolated organic fractions of methanolic leaf extract of *Azadirachta indica* significantly inhibited *Naja nigricollis* venom enzymes in vitro. *A. indica* extract acts as an effective adjuvant when used with snake venom antiserum in the treatment [169]. Aqueous and methanolic extracts of *Azadirachta indica* leaves are protective against *Naja haje arabica* and *Bitis arietans arietans* snake venoms in mice [170]. Leaf extract of *Azadirachta indica* has a clotting agent that acts against Russell’s viper venom [171]. The modified gedunin from *Azadirachta indica* showed improved pharmacological properties for combating snakebites in the molecular docking studies against snake venom enzymes [172]. A mixture of aqueous ethanolic extracts of *Azadirachta indica, Areca catechu, Butea monosperma, Citrus limon* and *Clerodendrum serratum* inhibited different components of *Naja naja, Bungarus caeruleus, Daboia russelii* and *Echis carinatus* venoms in different in vivo and in vitro studies [173].
Moraceae			
	*Ficus religiosa* L.	Leaves [174,175,176] Flowers [69]	None
Musaceae			
	*Musa acuminata* Colla *Musa x paradisiaca L.*	Plant * [177]	*Musa paradisiaca* extract when mixed with Crotalidae venoms in vitro and administered in mice, inhibited venom lethality, haemorrhagic activity, myotoxicity and phospholipase A2 [177,178].
Myrsinaceae			
	*Embelia ribes* Burm.f.	Seeds [179,180] Plant * [181,182]	None
Nyctaginaceae			
	*Boerhavia diffusa* L. Nom. Cons.	Roots [183] Leaves [184,185,186]	Ethanolic extract of *Boerhavia diffusa* inhibited phospholipase A2 in significant amounts [187].
Papaveraceae			
	*Papaver somniferum* L.	Plant * [69]	None
Pedaliaceae			
	*Sesamum indicum* L.	Seed oil [188]	None
Piperaceae			
	*Piper betle* L.	Whole Plant [189] Leaves [18]	None
	*Piper longum* L.	Roots [190] Roots, Fruits [191,192] Fruits [18] Fruits ^ [118]	Ethanolic extract of *Piper longum* fruits inhibited Russell’s viper venom-induced defibrinogenation, haemorrhage, inflammation, and lethality and proved to be a good antisnake venom [42]. Methanolic extract of *Piper longum* fruit neutralized procoagulant, proteolytic activity and phospholipase A2 from *Daboia russelii* venom when tested in vitro [44].
	*Piper nigrum* L.	Fruits [193,194] Fruits ^ [118] Fruits, Seeds [190,195] Seeds [30,105]	None
Plumbaginaceae			
	*Plumbago zeylanica* L.	Roots [196,197,198,199] Leaves [200]	None
Ranunculaceae			
	*Aconitum ferox* Wall. ex Ser.	Roots [194,201] Rhizomes [202]	The aqueous root extract of *Aconitum ferox* administered orally was evaluated in mice against *Naja naja* venom. No effect on venom mortality was reported; however, reversal of ill effects on Lymphocyte-Neutrophil Ratio and SGOT were observed [203].
	*Aconitum heterophyllum* Wall.	Tubers [204]	None
Rutaceae			
	*Aegle marmelos* (L.) Corr.	Whole plant [205,206,207] Roots [11] Roots ^ [118] Root bark [17] Stem bark [18] Leaves [208] Roots, Leaves, Stem bark [209,210,211]	Aqueous ethanolic extract of *Aegle marmelos* inhibited various enzymes from *Naja naja* venom when tested in vitro [74]. Ethanol, methanol and hexane extracts of leaves, stem bark and root bark of *Aegle marmelos* were tested by in vitro and in silico methods against *Naja naja* venom. Venom lethality inhibitory activity, antihaemolytic activity, antiacetylcholinesterase activity and antiproteolytic activity were presented by the extracts [212].
	*Citrus aurantifolia* (Christm.) Swingle	Fruit juice [213]	None
Salicaceae			
	*Casearia graveolens* Dalz.	Twigs [214] Plant * [215]	None
Sapindaceae			
	*Sapindus mukorossi* Gaertn. *Sapindus laurifolius* Vahl. *Sapindus trifoliatus* L.	Leaves, Stem bark [216] Fruits [217]	Methanolic extract of *Sapindus laurifolius* fruit neutralized procoagulant, proteolytic activity and phospholipase A2 from *Daboia russelii* venom when tested in vitro [44].
Solanaceae			
	*Nicotiana tabacum* L.	Roots, Leaves [218] Leaves [219,220] Seeds [18] Plant * [120,221,222,223]	None
Zingiberaceae			
	*Curcuma longa* L.	Rhizomes [11,69,224] Rhizomes ^ [118] Essential oil from rhizome [225]	Fractionated methanolic extract of *Curcuma longa* neutralized activity of 5′-nucleotidase from *Echis carinatus* venom by 83.7% when compared with snake venom antiserum [75]. Turmerin from *Curcuma longa* inhibited phospholipase A2 from *Naja naja* (cobra) snake venom and prevented cytotoxicity, oedema and myotoxicity [226]. Ar-Turmerone from *Curcuma longa* neutralized haemorrhagic activity from *Bothrops jararaca* venom and lethal effect of *Crotalus durissus terrificus* venom when tested in mice [227]. The binding of curcumin analogues with phospholipase A2 was tested via molecular docking studies, and dihydrocurucmin, tetrahydrocurcumin, hexahydrocurcumin and rosmarinic acid were found to have the more binding potential [228].
	*Elettaria cardamomum* L. Maton	Seeds [229,230,231] Seeds, Pods [232]	None
	*Zingiber officinale* Roscoe	Rhizomes [233,234,235] Rhizomes ^ [118] Rhizomes ~ [30]	None

^$^ Mention of the plant part in modern literature with a context of traditional or folk literature. * Specific plant part used in the therapy is not known. ^ As one of the parts of *Bilwadi Gutika*. ~ Used in combination with other plants/drugs.

**Table 5 tropicalmed-07-00108-t005:** Snake species-wise literature references indicating pharmacological action, enzyme inhibition and toxin inhibition.

Family		References		
	Snake Species	Action	Enzyme X	Toxin X
Elapidae				
	*Bungarus caeruleus*		[173]	[96]
	*Dendroaspis polylepis polylepis*			[96]
	*Naja haje arabica*	[170]		
	*Naja kaouthia*	[104,144]	[167]	
	*Naja naja*	[64,78,145,203,212,226]	[64,74,94,95,108,145,173,212]	[96,145,226]
	*Naja naja karachiensis*	[128]	[60,61,62]	
	*Naja nigricollis*	[168,169]	[169]	
	*Ophiophagus hannah*	[104]		
	*Oxyuranus microlepidotus*			[96]
Viperidae				
	*Bitis arietans*	[78,170]		
	*Bothrops jararaca*	[158,227]		
	*Calloselasma rhodostoma*	[113,116]		
	Crotalid species	[115,177,178]	[177,178]	[177,178]
	*Crotalus adamanteus*	[17]	[17]	
	*Crotalus durissus terrificus*	[114,227]		
	*Daboia russelii*	[42,44,104,106,107,144,145,156,171]	[44,77,104,107,109,110,145,156,173]	[145]
	*Echis carinatus*	[104]	[75,173]	
	*Echis ocellatus*	[78]		
	Unspecified snake species	[159]	[76,159,172,187,228]	

**Table 6 tropicalmed-07-00108-t006:** Plant species with no modern literature evidence as snake envenomation antidote.

Family	Plant Species
Asteraceae	*Anacyclus pyrethrum* (L.) Link.
	*Inula racemosa* Hook.f.
	*Launaea procumbens* L.
Burseraceae	*Garuga pinnata* Roxb.
Cactaceae	*Opuntia elatior* Mill.
Cucurbitaceae	*Luffa echinata* Roxb.
Euphorbiaceae	*Ricinus communis* L.
Malvaceae	*Hibiscus sabdariffa* L.
	*Thespesia populnea* (L.) Sol. ex Corrêa
Mimosaceae	*Albizia saman* (Jacq.) F.Muell.
Moraceae	*Ficus racemosa* L.
Myristicaceae	*Myristica fragrans* Houtt.
Oleaceae	*Jasminum coarctatum* Roxb.
Papaveraceae	*Fumaria indica* Lam.
Selaginellaceae	*Selaginella bryopteris* (L.) Baker
Solanaceae	*Hyoscyamus niger* L.
	*Solanum indicum* L.
Vitaceae	*Cissus quadrangularis* L.

## Data Availability

Not applicable.

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
