# Peer review of "A Contemporary Exploration of Traditional Indian Snake Envenomation Therapies"

_tropicalmed, 2022, doi:10.3390/tropicalmed7060108_

Round 1
Reviewer 1 Report
the manuscript satisfactorily explored the traditional therapies used to treat envenomation by snakes in India, correlating with the main plants involved. The review is potentially relevant for that population due to the high number of snake species involved in human envenomation, consequently, causing numerous and severe cases of envenomation per year in that magnific country.
The main subject of the manuscript is intensely argued and this reviewer would recommend its publication in the present form.
Author Response
The authors are thankful to the reviewer for their time to assess the manuscript and for encouraging feedback. Research on snake venom is in decline and the authors hope that this manuscript would revive some interest in readers’ minds to take up venom research.
Authors have incorporated changes as suggested by reviewers, thoroughly checked the manuscript for spelling and grammatical errors and the revised manuscript is being submitted.
Reviewer 2 Report
The manuscript brings us a rescue of the medical literature of traditional practitioners using plants for treating snakebites, based on two classic Ayurveda books from India. Also, a vast list of plants is evaluated by their potential pharmacological effects on venoms or toxins of several snakes, pointing out possible active agents in neutralizing the effects of those envenoming.
Unfortunately, the description of the ancient methods for treating snakebites was very disappointing, of the formulation used and the routes of administration. Some of them seem to be potentially dangerous for the patients.
Another point is that most plant species listed in Table 4 have no pharmacological evidence of action. Emphasizing the ones with some pharmacological action described and discussing the possible effect of those plants with no description of pharmacological effects could be more attractive.
Also, figures 2 and 3 show the number of references evidencing plants with inhibitory effects on snake venoms and their components. A table with the name of the snake species and a list of the references evidencing the effect of plant or plant components on a venom, toxin, or enzyme could be more helpful.
It is an interesting review article but could emphasize less the ancient non-effective practices and more the potential effect of the plants described as helpful in treating snakebites.
Author Response
The authors are thankful to the reviewer for their keen interest in making this manuscript better and more useful to the readers. The authors would like to discuss the following points based on the feedback.
Point 1: the description of the ancient methods for treating snakebites was very disappointing, of the formulation used and the routes of administration. Some of them seem to be potentially dangerous for the patients.
Discussion 1: Most of the enlisted therapies for treating snakebites are from compilations of traditional healers' personal journals based upon experiences. Authors have tried to maintain the original meaning and presentation of treatment methods while translating from vernacular language. Quick worsening of snakebite victim's physiological condition post envenomation and non-availability of definitive antidote might have led practitioners to try diabolical remedies in the earlier days. Despite that, authors have enlisted such (potentially dangerous and probably non-effective) therapies, 1. To include the plant species for consideration as an antidote and 2. To create awareness in the present day to eliminate such practices (possibly) conducted in remote tribal areas. The related points are discussed in section 3.2 ‘Therapeutic anomalies’ and section 12 of the manuscript.
Point 2: most plant species listed in Table 4 have no pharmacological evidence of action. Emphasizing the ones with some pharmacological action described and discussing the possible effect of those plants with no description of pharmacological effects could be more attractive.
Discussion 2: Discussion on plants with no known pharmacological effects would be really helpful in the current scenario when only a few plant species are investigated and the majority are not. This would be great and conclusive when in-depth thought is given to phytoconstituents for their known effects and discussed accordingly. As the number of uninvestigated plants is relatively high (30 from table 4 and 18 from table 6) authors would now take this up as a separate project to conduct. To avoid further lengthening of the current article, a paragraph hinting at further investigation has been added in section 12 of the manuscript.
Point 3: figures 2 and 3 show the number of references evidencing plants with inhibitory effects on snake venoms and their components. A table with the name of the snake species and a list of the references evidencing the effect of plant or plant components on a venom, toxin, or enzyme could be more helpful.
Discussion 3: The table would be a ready reference to the readers and would increase utility. A compiled table is added (table 5) to the manuscript. As one referred article contains more than one plant investigated, the numbers of appearances in the chart are higher than the number of articles cited.
It is encouraging to receive positive remarks and critical comments to improve the manuscript. The authors thank the reviewer for their time and encouragement. Authors have incorporated changes as suggested by reviewers, thoroughly checked the manuscript for spelling and grammatical errors and the revised manuscript is being submitted.
Reviewer 3 Report
This article deals with the review of anti-snake venom phytotherapy from Indian traditional medicine. It brings a compilation of several species of plants, plant parts, and mixtures that have been used to deter snake envenomation. Despite most plant-based therapies mentioned in the traditional or modern literature lacking clinical evidence, they deserve attention for harnessing natural products with anti-snake venom properties. Thus, this manuscript could be a source of information for further developing natural substances with clinical evidence that will serve for snake envenomation and their symptoms.
Author Response
The authors are thankful to the reviewer for their encouraging comments and time to assess the manuscript. This encouragement would lead authors to contribute more to the research related to snake venom.
Authors have incorporated changes as suggested by reviewers, thoroughly checked the manuscript for spelling and grammatical errors and the revised manuscript is being submitted.